# Identification of Antitumor *miR-30e-5p* Controlled Genes; Diagnostic and Prognostic Biomarkers for Head and Neck Squamous Cell Carcinoma

**DOI:** 10.3390/genes13071225

**Published:** 2022-07-09

**Authors:** Chikashi Minemura, Shunichi Asai, Ayaka Koma, Naoko Kikkawa, Mayuko Kato, Atsushi Kasamatsu, Katsuhiro Uzawa, Toyoyuki Hanazawa, Naohiko Seki

**Affiliations:** 1Department of Oral Science, Graduate School of Medicine, Chiba University, Chiba 260-8670, Japan; minemura@chiba-u.jp (C.M.); axna4812@chiba-u.jp (A.K.); kasamatsua@faculty.chiba-u.jp (A.K.); uzawak@faculty.chiba-u.jp (K.U.); 2Department of Functional Genomics, Chiba University Graduate School of Medicine, Chiba 260-8670, Japan; cada5015@chiba-u.jp (S.A.); naoko-k@hospital.chiba-u.jp (N.K.); mayukokato@chiba-u.jp (M.K.); 3Department of Otorhinolaryngology/Head and Neck Surgery, Chiba University Graduate School of Medicine, Chiba 260-8670, Japan; thanazawa@faculty.chiba-u.jp

**Keywords:** microRNA, HNSCC, tumor-suppressor, *miR-30e-5p*, *FOXD1*, TCGA

## Abstract

Analysis of microRNA (miRNA) expression signatures in head and neck squamous cell carcinoma (HNSCC) has revealed that the *miR-30* family is frequently downregulated in cancer tissues. The Cancer Genome Atlas (TCGA) database confirms that all members of the *miR-30* family (except *miR-30c-5p*) are downregulated in HNSCC tissues. Moreover, low expression of *miR-30e-5p* and *miR-30c-1-3p* significantly predicts shorter survival of HNSCC patients (*p* = 0.0081 and *p* = 0.0224, respectively). In this study, we focused on *miR-30e-5p* to investigate its tumor-suppressive roles and its control of oncogenic genes in HNSCC cells. Transient expression of *miR-30e-5p* significantly attenuated cancer cell migration and invasive abilities in HNSCC cells. Nine genes (*DDIT4*, *FOXD1*, *FXR1*, *FZD2*, *HMGB3*, *MINPP1*, *PAWR*, *PFN2*, and *RTN4R*) were identified as putative targets of *miR-30e-5p* control. Their expression levels significantly predicted shorter survival of HNSCC patients (*p* < 0.05). Among those targets, *FOXD1* expression appeared to be an independent factor predicting patient survival according to multivariate Cox regression analysis (*p* = 0.049). Knockdown assays using siRNAs corresponding to *FOXD1* showed that malignant phenotypes (e.g., cell proliferation, migration, and invasive abilities) of HNSCC cells were significantly suppressed. Overexpression of *FOXD1* was confirmed by immunostaining of HNSCC clinical specimens. Our miRNA-based approach is an effective strategy for the identification of prognostic markers and therapeutic target molecules in HNSCC. Moreover, these findings led to insights into the molecular pathogenesis of HNSCC.

## 1. Introduction

A head and neck squamous cell carcinoma (HNSCC) may arise in the oral cavity, hypopharynx, nasopharynx, and larynx. Based on Global Cancer Statistics 2018, HNSCC is the eighth most common cancer in the world [1]. Approximately 890,000 people are diagnosed with HNSCC annually, and 450,000 die from the disease [2]. A notable characteristic of HNSCC is that many patients are already in the advanced stage of disease at the time of diagnosis. Treatment strategies are limited for patients at such an advanced stage [3]. Cisplatin-based treatment is selected for patients for whom surgery is not indicated. However, cancer cells develop drug-resistance in the course of treatment [4]. No effective treatments have been reported for patients with HNSCC who have acquired drug resistance [5]. Thus, there is a pressing need for improved understanding of the molecular pathogenesis of HNSCC using the latest genomic science.

The human genome project showed that a vast number of non-coding RNA molecules (ncRNAs) are transcribed and function in normal and diseased cells [6,7]. MicroRNAs (miRNAs), which constitute a class of ncRNA, are single strands of RNA, 19 to 23 bases in length. The unique nature of miRNA is that a single miRNA negatively controls a large number of RNA transcripts (both protein coding RNAs and ncRNAs) in a sequence-dependent manner [8,9]. Therefore, dysregulated miRNA expression can disrupt tightly controlled RNA networks in normal cells. The involvement of aberrantly expressed miRNAs has been reported in a variety of human diseases, including cancer [10].

Over the past decade, a large number of studies have shown that aberrantly expressed miRNAs are closely connected to human oncogenesis [11]. Identifying miRNAs with altered expression in cancer cells is a first step in the characterization of their role. The latest RNA-sequence technology is accelerating the creation of miRNA expression signatures in a variety of cancer types [12]. We have established miRNA expression signatures in several types of cancers, including HNSCC [13,14].

The analysis of miRNA signatures has revealed that members of the *miR-30* family are frequently downregulated in several types of cancers, including HNSCC [15,16]. In the human genome, the *miR-30* family is composed of five species, i.e., *miR-30a* to *miR-30e*. *miR-30c* is further subdivided into *miR-30c-1* and *miR-30c-2*. The *miR-30* family is encoded by six genes located on human chromosomes 1, 6, and 8 [17]. The guide strands of the *miR-30* family share the same seed sequence (GUAAACA). The passenger strands of the *miR-30* family are divided into two groups according to their seed sequences. The first group (UUUCAGU) includes *miR-30a-3p*, *miR-30d-3p*, and *miR-30e-3p*. The second group (UGGGAG) encompasses *miR-30b-3p*, *miR-30c-1-3p*, and *miR-30c-2-3p*. We theorized that clarifying the targets/pathways controlled by miRNAs for each cancer type could greatly enhance our understanding of the molecular pathogenesis of cancer cells.

The Cancer Genome Atlas (TCGA, https://www.cancer.gov/tcga), a landmark cancer genomics program, molecularly characterizes over 20,000 primary cancers (and matched normal samples) spanning 33 cancer types. Based on a TCGA analysis, all members of the *miR-30* family (except *miR-30c-5p*) are downregulated in HNSCC tissues. Moreover, the low expression of two miRNAs, *miR-30e-5p* and *miR-30c-1-3p*, significantly predicts shorter survival of patients with HNSCC (*p* = 0.0081 and *p* = 0.0224, respectively).

In this study, we focused on *miR-30e-5p* and investigated its control of genes involved in the molecular pathogenesis of HNSCC. A total of nine genes (*DDIT4*, *FOXD1*, *FXR1*, *FZD2*, *HMGB3*, *MINPP1*, *PAWR*, *PFN2*, and *RTN4R*) were identified as putative targets of *miR-30e-5p*. Their expression level predicted shorter survival of the patients (*p* < 0.05).

Among these genes, we focused on *FOXD1* (Forkhead Box D1), which belongs to the forkhead family of transcription factors. Here, we investigated its functional significance in HNSCC cells.

RNA immunoprecipitation (RIP) assays revealed that *FOXD1* mRNA was incorporated into the RNA-induced silencing complex (RISC). Additionally, dual luciferase reporter assays showed that *miR-30e-5p* was directly bound to the 3′-UTR of *FOXD1*. Functional assays using siRNAs targeting *FOXD1* showed that the knockdown of *FOXD1* expression attenuated HNSCC cell malignant behaviors (migration and invasion). Our approach will reveal the new insights into the involvement of miRNA and the molecular pathogenesis of HNSCC.

## 2. Materials and Methods

### 2.1. Analysis of miRNAs and miRNA Target Genes in HNSCC Patients

The sequences of the *miR-30* family were confirmed using miRbase ver. 22.1 (https://www.mirbase.org, accessed on 10 July 2020) [18]. TCGA-HNSC (TCGA, Firehose Legacy) was used to investigate the clinical significance of miRNAs and their target genes. Clinical parameters and gene expression data were obtained from cBioPortal (http://www.cbioportal.org/, data downloaded on 20 August 2020) and OncoLnc (http://www.oncolnc.org/, data downloaded on 20 August 2020) [19,20,21].

The putative target genes with the *miR-30-5p* binding sites were selected by using TargetScanHuman ver. 7.2 (http://www.targetscan.org/vert_72/, accessed on 22 November 2020) [22].

To identify differentially expressed genes in HNSCC tissues, a newly created microarray data (accession number: GSE180077) was used in this study. Three hypopharyngeal squamous cell carcinoma (HSCC) tissues, three normal hypopharyngeal tissues, and two cervical lymph nodes harvested from one HSCC patient who underwent surgical resection at Chiba University Hospital were subjected to Agilent whole genome microarrays. In this study, we compared gene expressions in cancer tissues with those in normal tissues. The clinical information of this patient was summarized in Appendix A.

### 2.2. HNSCC Cell Lines

Two human HNSCC cell lines were obtained from the RIKEN BioResource Center (Tsukuba, Ibaraki, Japan). Sa3 was harvested from upper gingiva cancer of 63y male. SAS was harvested from oral tongue cancer of 69y female.

### 2.3. RNA Extraction and Quantitative Real-Time Reverse-Transcription PCR (qRT-PCR)

Total RNA was isolated using TRIzol reagent (Invitrogen, Waltham, MA, USA) according to the manufacturer’s protocol. qRT-PCR was performed using the High-Capacity cDNA Reverse Transcription Kit (Applied Biosystems, Waltham, MA, USA) and the StepOnePlus™ Real-Time PCR System (Applied Biosystems). Gene expressions were quantified relatively by the delta-delta Ct method (used *GAPDH* as internal control). Taqman assays used in this study are summarized in Appendix A.

### 2.4. Transfection of miRNAs and siRNAs into HNSCC Cells

The protocols used for transient transfection of miRNAs and siRNAs were described in our previous studies [23,24,25,26]. The miRNA precursors and siRNAs used in this report were detailed in Appendix A. The miRNAs and siRNAs were transfected into HNSCC cell lines using Opti-MEM (Gibco, Carlsbad, CA, USA) and Lipofectamine^TM^ RNAiMax Transfection Reagent (Invitrogen, Waltham, MA, USA). All miRNA precursors were transfected into HNSCC cell lines at 10 nM, and siRNAs were transfected at 5 nM. Mock transfection consisted of cells without precursors or siRNAs. Control groups were transfected with the negative control precursor.

### 2.5. RIP Assay

The procedures for the RIP assay have been described previously [23]. Briefly, SAS cells were cultured in 10-cm plates at 80% confluency. Negative control miRNA precursors and *miR-30e-5p* precursors were transfected. After 6 h, immunoprecipitation was performed using the MagCapture^TM^ microRNA Isolation Kit, Human Ago2, obtained from FUJIFILM Wako Pure Chemical Corporation (Wako, Osaka, Japan) according to the manufacturer’s protocol. Expression levels of *FOXD1* bound to Ago2 were measured by qRT-PCR. Taqman primers used in the assay are summarized in Appendix A.

### 2.6. Functional Assays of HNSCC Cells (Cell Proliferation, Migration, and Invasion Assays)

The procedures for conducting functional assays (cell proliferation, migration, and invasion assays) with HNSCC cells were described in earlier publications [23,24,25,26]. In brief, for proliferation assays, Sa3 or SAS cells were transferred to 96-well plates at 3.0 × 10^3^ cells per well. Cell proliferation was evaluated using the XTT assay kit II (Sigma–Aldrich, St. Louis, MO, USA) 72 h after the transfection procedure. For migration and invasion assays, Sa3 and SAS cells were transfected in 6-well plates at 2.0 × 10^5^ cells per well. After 48 h, transfected Sa3 and SAS cells were added into each chamber at 1.0 × 10^5^ per well. Corning BioCoat^TM^ cell culture chambers (Corning, Corning, NY, USA) were used for migration assays whereas Corning BioCoat Matrigel Invasion Chambers were used for invasion assays. After 48 h, the cells on the lower surface of chamber membranes were stained and counted for analysis. All experiments were performed in triplicate.

### 2.7. Plasmid Construction and Dual-Luciferase Reporter Assays

The procedures for plasmid construction and the dual-luciferase reporter assays were outlined previously [23,24,25,26]. Briefly, a partial wild-type sequence, including the seed sequence of *FOXD1* within the 3′-untranslated region (3′-UTR), was inserted into the psiCHECK-2 vector (C8021; Promega, Madison, WI, USA). Alternatively, a deletion type that was missing the *miR-30e-5p* target site was synthesized. These synthesized vectors (50 ng) were transfected into 1.0 × 10^5^ cells in each well using Lipofectamine 2000 (Invitrogen, Waltham, MA, USA) and Opti-MEM. After 48 h of transfection, dual luciferase reporter assays using the Dual Luciferase Reporter Assay System (Promega, Madison, WI, USA) were conducted. Luminescence data were presented as the *Renilla/Firefly* luciferase activity ratio.

### 2.8. Immunohistochemistry

The procedures for immunohistochemistry were described in our previous studies [23,24,25,26]. The clinical samples were obtained from HNSCC cases who received surgical treatment at Chiba University Hospital. The slides were incubated with primary antibody FOXD1 (1: 50, PA5-27142, Thermo Fisher Scientific, Waltham, MA, USA). The clinical features are shown in Appendix A. The reagents used in the analysis are listed in Appendix A.

### 2.9. Gene Set Enrichment Analysis (GSEA)

To analyze the molecular pathways related to *FOXD1* (regulated by *miR-30e-5p*), GSEA was performed. Using TCGA-HNSC data, HNSCC patients were divided into high and low expression groups according to the Z-score of the *FOXD1* expression level. A ranked list of genes was generated by the log_2_ ratio comparing the expression levels of each gene between the two groups. The ranked gene list was uploaded into GSEA software [27,28] and we applied the Hallmark gene set in The Molecular Signatures Database [27,29].

### 2.10. Statistical Analysis

Statistical analyses were determined using JMP Pro 15 (SAS Institute Inc., Cary, NC, USA). Welch’s *t*-tests were performed to determine the significance of differences between two groups. Dunnett’s tests were applied for comparisons among multiple groups. For correlation analyses, Spearman’s test was applied. Survival analyses were analyzed by log-rank test. Each expression levels of target genes, age, disease stage, and pathological grade in TCGA-HNSC were used as variables for Cox’s proportional hazards model. A *p*-value less than 0.05 was considered statistically significant.

## 3. Results

### 3.1. Expression Levels and the Clinical Significance of the miR-30 Family in HNSCC Clinical Specimens Assessed by TCGA Analysis

The human *miR-30* family consists of 12 members. They include *miR-30a-5p*, *miR-30a-3p*, *miR-30b-5p*, *miR-30b-3p*, *miR-30c-1-5p*, *miR-30c-1-3p*, *miR-30c-2-5p*, *miR-30c-2-3p*, *miR-30d-5p*, *miR-30d-3p*, *miR-30e-5p*, and *miR-30e-3p*. Seed sequences of *miR-30-5p* are identical. In contrast, two types of seed sequences are present in *miR-30-3p* (Figure 1A). Human *miR-30e* and *miR-30c-1* are located on chromosome 1p34.2, whereas *miR-30c-2* and *miR-30a* are on chromosome 6q13, and *miR-30b* and *miR-30d* are found on chromosome 8q24.22 (Figure 1B).

Expression levels of all members of the *miR-30* family were validated by TCGA database analyses. All members of the *miR-30* family (except for *miR-30c-5p*) were significantly downregulated in HNSCC tissues (*n* = 484) compared to normal tissues (*n* = 44) (Figure 2).

Moreover, low expression of *miR-30e-5p* and *miR-30c-1-3p* significantly predicted poor prognosis of patients with HNSCC (Figure 3). Expression levels of other members of the *miR-30* family were not related to patient prognosis (Figure 3).

### 3.2. Effect of Transient Transfection of miR-30e-5p on HNSCC Cell Proliferation, Migration and Invasion

In this study, we focused on *miR-30e-5p* because its expression was significantly downregulated in HNSCC tissues, and it was closely associated with poor prognosis of the patients, suggesting that *miR-30e-5p* acts as a tumor-suppressive miRNA in HNSCC cells.

The tumor-suppressive roles of *miR-30e-5p* were assessed by transient transfection of *miR-30e-5p* in two cell lines, Sa3 and SAS. Transient transfection of *miR-30e-5p* inhibited HNSCC cell proliferation (Figure 4A). Cancer cell migration and invasive abilities were markedly blocked by *miR-30e-5p* expression in Sa3 and SAS cells (Figure 4B,C). Photographs of typical results from the migration and invasion assays are shown in Appendix A.

### 3.3. Screening for Oncogenic Targets of miR-30e-5p in HNSCC

To identify genes modulated by *miR-30e-5p* that were closely involved in HNSCC molecular pathogenesis, we performed in silico database analyses combined with our gene expression data. We created new gene expression data using clinical specimens of hypopharyngeal squamous cell carcinoma (HSCC). Three HSCC tissues, three normal hypopharyngeal tissues, and two cervical lymph nodes harvested from one HSCC patient who underwent surgical resection at Chiba University Hospital were subjected to Agilent whole genome microarrays. In this study, we compared gene expressions in cancer tissues with those in normal tissues. The clinical information of this patient was summarized in Appendix A. Expression data were deposited in the GEO database (accession number: GSE180077). Our selection strategy is shown in Figure 5.

TargetScan Human database (release 7.2) provides data on the putative targets of *miR-30e-5p*. A total of 1576 genes are listed. Among these targets, 97 genes were upregulated in HSCC patients. Furthermore, the increased expression of 54 genes was confirmed in TCGA database analyses (Table 1).

### 3.4. Clinical Significance of miR-30e-5p Targets in Patients with HNSCC Determined by TCGA Analysis

Among the 54 putative target genes, high expression of nine genes (*DDIT4*, *FOXD1*, *FXR1*, *FZD2*, *HMGB3*, *MINPP1*, *PAWR*, *PFN2*, and *RTN4R*) showed statistically significant correlations with the 5-year overall survival frequencies of patients with HNSCC (*p* < 0.05; Figure 6 and Figure 7).

Therefore, univariate analysis for 5-year overall survival was conducted first, and then multivariate analysis was performed for the statistically significant variables (*p* < 0.05). Each expression levels of target genes, age, disease stage, and pathological grade in TCGA-HNSC were used as variables for Cox’s proportional hazards model. As a result, the high expression level of *FOXD1* (HR: 1.374, 95% CI: 1.002–1.890, *p* = 0.049), age (≥70) (HR: 1.922, 95% CI: 1.369–2.698, *p* < 0.001) and disease stage (III and IV) (HR: 1.774, 95% CI: 1.159–2.716, *p* = 0.008) were independent prognostic factors (Table 2).

The correlations of the expression levels between these nine genes and *miR-30e-5p* were evaluated by TCGA-HNSC. A Spearman’s rank test confirmed weak negative correlations in seven genes (*DDIT4*, *FOXD1*, *FZD2*, *MINPP1*, *PAWR*, *PFN2*, and *RTN4R*) (*p* < 0.001, *r* < −0.2, Figure 8), whereas the correlations were not confirmed in *FXR1* and *HMGB3*.

### 3.5. Regulated Expression of the Nine Identified Genes by miR-30e-5p in HNSCC Cells

qRT-PCR revealed that the mRNA expression levels of *DDIT4*, *FOXD1*, *MINPP1*, *PAWR*, and *PFN2* were significantly suppressed in *miR-30e-5p*-transfected HNSCC cells. The other genes were not significantly suppressed by *miR-30e-5p* (Figure 9).

Considering the multivariate analysis, the correlation analysis, and the results of qRT-PCR, *FOXD1* was of the greatest interest among the nine targets. Thus, we focused on *FOXD1* as the target gene of *miR-30e-5p* in this study.

### 3.6. Incorporation of FOXD1 mRNA into the RNA-Induced Silencing Complex (RISC) and Direct Control of FOXD1 Expression by miR-30e-5p in HNSCC Cells

To confirm incorporation of *FOXD1* mRNA into RISC, RIP assays were performed (Figure 10A,B). The schematic illustration displays the concept of RIP assays. Ago2-bound miRNA and mRNA were isolated by the immunoprecipitation of Ago2, the protein that plays a central role in RISC (Figure 10A). Using samples isolated by immunoprecipitation, qRT-PCR showed that the *FOXD1* mRNA level was significantly higher than those of mock or miR control transfected cells (*p* < 0.01; Figure 10B), suggesting a significant incorporation into RISC.

To confirm that *miR-30e-5p* bound directly to the 3′-UTR of *FOXD1*, a dual-luciferase reporter assay was performed. Luciferase activity was significantly reduced following co-transfection with *miR-30e-5p* and a vector containing the *miR-30e-5p*-binding site in the 3′-UTR of *FOXD1* (Figure 10C). In contrast, co-transfection with a vector containing the *FOXD1* 3′-UTR, in which the *miR-30e-5p*-binding site was deleted, resulted in no change in luciferase activity (Figure 10C).

### 3.7. Expression of FOXD1 in HNSCC Clinical Specimens

HNSCC clinical specimens displayed moderate immunoreactivity in the cytoplasm (Figure 11B,D), whereas normal epithelium showed no expression of *FOXD1* (Figure 11A,C). The clinical features of HNSCC specimens are summarized in Appendix A.

### 3.8. Effects of FOXD1 Knockdown on the Proliferation, Migration, and Invasion of HNSCC Cells

To assess the tumor-promoting effect of *FOXD1* in HNSCC cells, we performed knockdown assays using siRNAs.

First, the inhibitory effects of two different siRNAs targeting *FOXD1* (si*FOXD1*-1 and si*FOXD1*-2) expression were examined. The *FOXD1* mRNA levels were effectively inhibited by each siRNA (Appendix A).

The knockdown of *FOXD1* slightly inhibited Sa3 and SAS cell proliferation (Figure 12A). In contrast, cell migration and invasion were significantly inhibited after si*FOXD1*-1 and si*FOXD1*-2 transfection into Sa3 and SAS cells (Figure 12B,C). Photographs of typical results from the migration and invasion assays are shown in Appendix A.

### 3.9. FOXD1-Mediated Molecular Pathways in HNSCC Cells

We performed gene set enrichment analysis (GSEA) to identify genes that were differentially expressed in the high and low expression groups using TCGA-HNSCC data. A GSEA analysis of *FOXD1* showed that the most enriched molecular pathway in the high expression groups of *FOXD1* was “epithelial-mesenchymal transition” (Table 3, Figure 13). Additional pathways (MYC targets, TNFα signaling, Hypoxia, E2F targets, Glycolysis, and DNA repair) were also associated with groups expressing high levels of *FOXD1* (Table 3). The aberrant expression and activation of these pathways were closely associated with the downregulation of *miR-30e-5p* in HNSCC cells and contributed to HNSCC oncogenesis.

## 4. Discussion

In the human genome, numerous miRNAs have very similar sequences and therefore constitute a “family”. Because the seed sequences of the miRNA family are the same, each miRNA family controls the expression of the same set of genes. Therefore, aberrant expression of the miRNA family members will cause the disruption of intracellular molecular networks, and these events can initiate the transformation of normal cells to cancer cells. Based on our miRNA signatures, we previously focused on several miRNA families and identified the molecular pathways that were controlled by the *miR-29* family, the *miR-199* family and the *miR-216* family [14,23,30].

Our RNA sequence-based miRNA signatures, including those in HNSCC, showed that some *miR-30* family members were frequently downregulated in cancer tissues. Those observations suggest that these miRNAs control targets with pivotal roles in cancer progression, metastasis, and drug-resistance [31]. Here, we first investigated the expression levels and clinical significance of all members of the *miR-30* family using TCGA database. The family consists of 12 miRNAs: guide and passenger strands of *miR-30a*, *miR-30b*, *miR-30c-1*, *miR-30c-2*, *miR-30d*, and *miR-30e*). Among them, the expression levels of *miR-30e-5p* and *miR-30c-1-3p* were closely associated with HNSCC molecular pathogenesis. Identifying molecular networks controlled by these miRNAs is important for understanding HNSCC oncogenesis.

In this study, we focused on *miR-30e-5p* to investigate the targets that it controls in HNSCC. The seed sequences of the guide strands of the *miR-30a* family are identical, suggesting that these miRNAs may control common targets in a sequence-dependent manner. Previous studies reported that *miR-30e-5p* had tumor-suppressive functions in several types of cancers [32,33,34]. A previous study showed that *miR-30e-5p* suppressed astrocyte elevated gene-1 (*AEG-1*), an oncogene that contributes to angiogenesis, metastasis, and EMT processes [35]. Another study showed that *miR-30e-5p* was a direct transcriptional target of P53 in colorectal cancer. Expression of *miR-30e-5p* blocked tumor cell migration, invasion, and in vivo metastasis by directly controlling integrin molecules [36].

A single miRNA controls a large number of genes, and targeted genes can differ depending on the type of cancer. Defining miRNA-targeted genes is an important focus in miRNA research. Our target identification strategy successfully identified nine genes that significantly predicted 5-year survival frequencies of HNSCC patients. Unfortunately, none of those genes were independent prognostic factors contributing to 5-year overall survival rates. According to multivariate analysis, a high expression level of *FOXD1* was an independent prognostic factor (HR: 1.374, 95% CI: 1.002–1.890, *p* = 0.049). Furthermore, the negative correlation between the expression of *FOXD1* and *miR-30e-5p* was confirmed. For those reasons, we focused on *FOXD1* as the target gene in this study. Future studies may examine the other genes in the belief that resultant data will improve our understandings of the molecular pathogenesis of HNSCC.

For example, *FXR1* is an RNA-binding protein that regulates co-transcriptional and post-transcriptional gene expression. It is a member of the Fragile X-mental retardation (*FXR*) family of proteins, which includes *FMR1* and *FXR2* [37]. Recent studies showed that the overexpression of *FRX1* was observed in several types of cancers, including HNSCC, and its expression correlates with poor prognosis of the patients [38]. In oral cancer cells, *FXR1* stabilized *miR-301a-3p* and *miR-301a-3p*, both of which target *p21* [39]. Another study showed that the overexpression of *FXR1* facilitates the bypass of senescence and tumor progression [40]. Aberrant expression of *FXR1*-enhanced HNSCC progression and its expression is closely associated with the molecular pathogenesis of HNSCC.

*PFN2* was identified as a *miR-30e-5p* target in this study. *PFN2* is a member of the profilin family, namely, profilin 1, 2, 3, and 4 [41]. Profilins are actin-binding proteins involved in the regulation of cytoskeletal dynamics. Overexpression of *PFN2* was reported in several types of cancers, including HNSCC [42]. Recently, we revealed that *PFN2* was directly regulated by tumor-suppressive *miR-1*/*miR-133* clustered miRNAs in HNSCC cells, and its overexpression promoted cancer cell malignant transformation [25]. Another study showed that overexpression of *PFN2* enhanced the aggressiveness of small cell lung cancer (SCLC), including cell proliferation, migration, and invasion. Knockdown of *PFN2* decreased the cells’ aggressive nature. Moreover, a mouse xenograft model demonstrated that the overexpression of *PFN2* dramatically elevated SCLC growth and vasculature formation as well as lung metastasis [43]. Notably, a previous study showed that *miR-30a-5p* suppressed the expression of *PFN2* in lung cancer cell lines [44].

In this study, we confirmed that the overexpression of *FOXD1* facilitated cancer cell malignant transformation, e.g., enhanced migratory and invasive abilities in HNSCC cells. *FOXD1* is a member of the forkhead family of transcription factors [45]. *FOXD1* expression is upregulated in several types of cancers [46]. Moreover, the knockdown of *FOXD1* impairs the colony-forming abilities of oral cancer cells after radiation treatment [47]. Another study showed that upregulated *FOXD1* contributed to melanoma cells’ resistance to vemurafenib via the recruited expression of connective tissue growth factor [48]. In gastric cancer (GC) cells, *FOXD1*-AS1 expression induced *FOXD1* translation, and these events enhanced GC cell progression and cisplatin resistance [49]. Importantly, *miR-30a-5p* directly binds to the 3′-UTR of *FOXD1* and suppresses its expression in several types of cancer cells, e.g., lung squamous cell carcinoma, pancreatic ductal adenocarcinoma, osteosarcoma, and ovarian cancer [50,51,52,53]. Our present data and previous studies indicate that the overexpression of *FOXD1* is closely involved in aggressive cancer cell transformation in a wide range of cancers, including HNSCC.

These studies show that searching for target genes of tumor-suppressive miRNAs is an attractive strategy for exploring the molecular mechanisms of HNSCC.

## 5. Conclusions

We showed that the *FOXD1* gene is directly controlled by tumor-suppressive *miR-30e-5p* in HNSCC cells. The aberrant expression of *FOXD1* facilitated cancer cell migration and invasion, and its expression was closely associated with the prognosis of HNSCC patients. Our strategy (analysis of tumor-suppressive miRNAs and their controlled genes) can identify genes that are deeply involved in the molecular pathogenesis of HNSCC.

## Figures and Tables

**Figure 1 genes-13-01225-f001:**
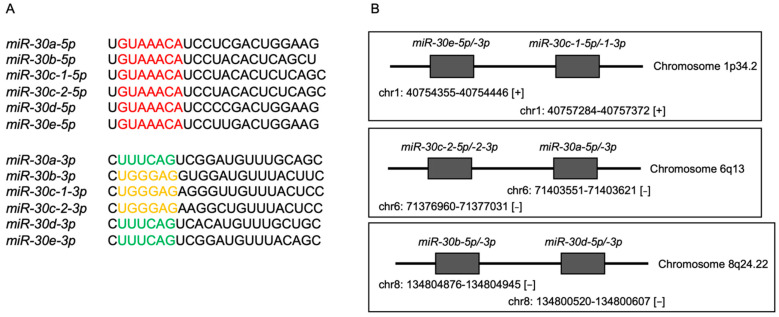
Twelve miRNAs are present in the human genome as members of the *miR-30* family. (**A**) Seed sequences of *miR-30-5p* are identical. In contrast, two types of seed sequences are present in *miR-30-3p*. (**B**) The locations of each microRNA family on chromosome.

**Figure 2 genes-13-01225-f002:**
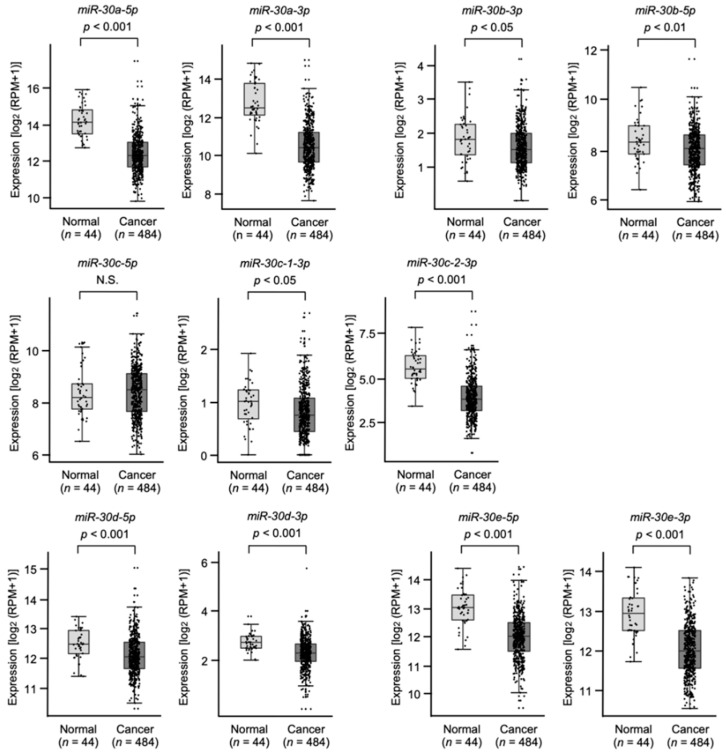
The expression level of the *miR-30* family was analyzed using TCGA-HNSC database. A total of 484 HNSCC tissues and 44 normal epithelial tissues were evaluated (N.S.: not significant).

**Figure 3 genes-13-01225-f003:**
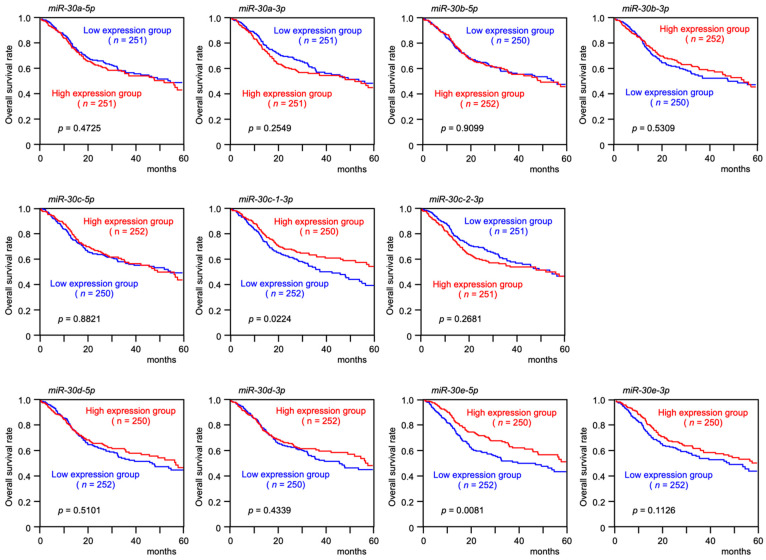
Kaplan–Meier survival analyses of HNSCC patients using data from TCGA-HNSC. Patients were divided into two groups according to the median miRNA expression level: high and low expression groups. The red and blue lines represent the high and low expression groups, respectively.

**Figure 4 genes-13-01225-f004:**
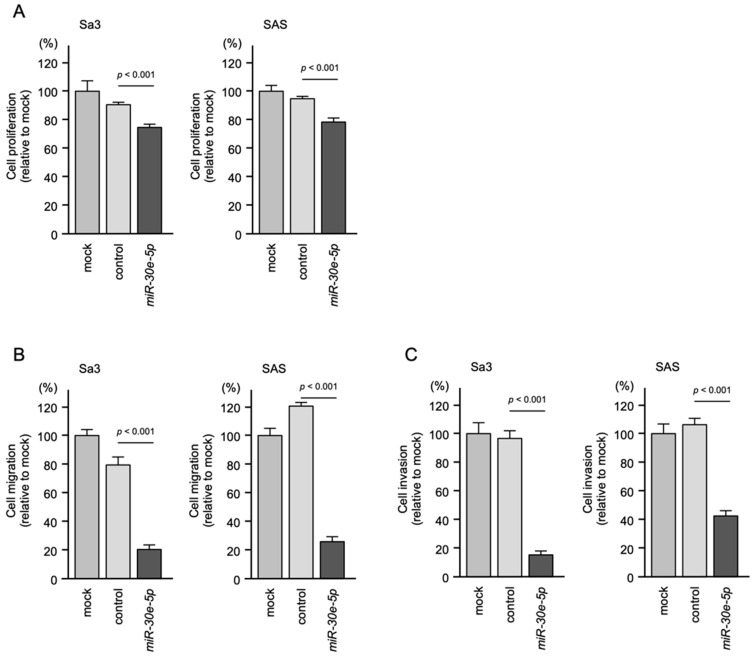
Functional assays of *miR-30e-5p* in HNSCC cell lines (Sa3 and SAS)**.** (**A**) Cell proliferation was assessed using XTT assays 72 h after miRNA transfection. (**B**) Cell migration was assessed using a membrane culture system 48 h after seeding miRNA-transfected cells into the chambers. (**C**) Cell invasion was determined using Matrigel invasion assays 48 h after seeding miRNA-transfected cells into the chambers.

**Figure 5 genes-13-01225-f005:**
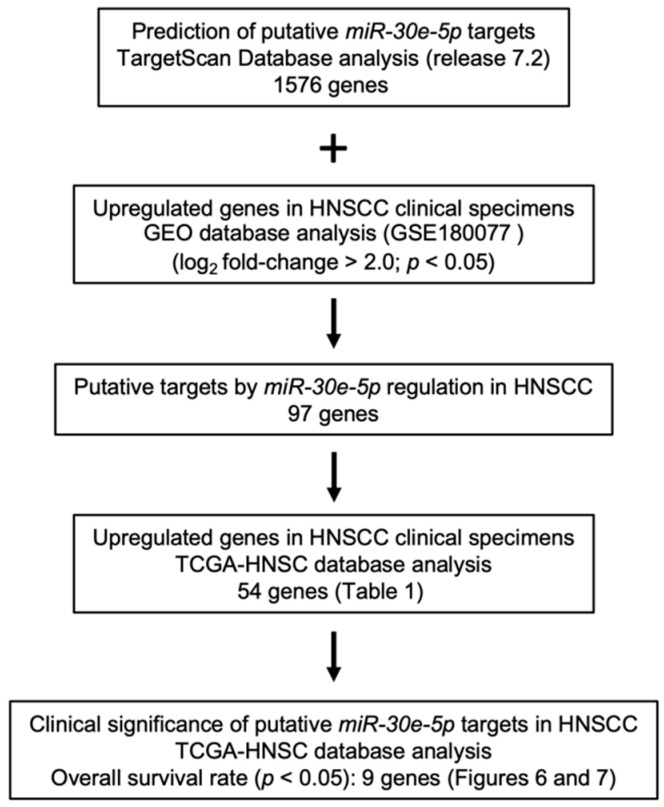
Flow chart of the strategy used to identify putative tumor suppressor genes regulated by *miR-30e-5p*.

**Figure 6 genes-13-01225-f006:**
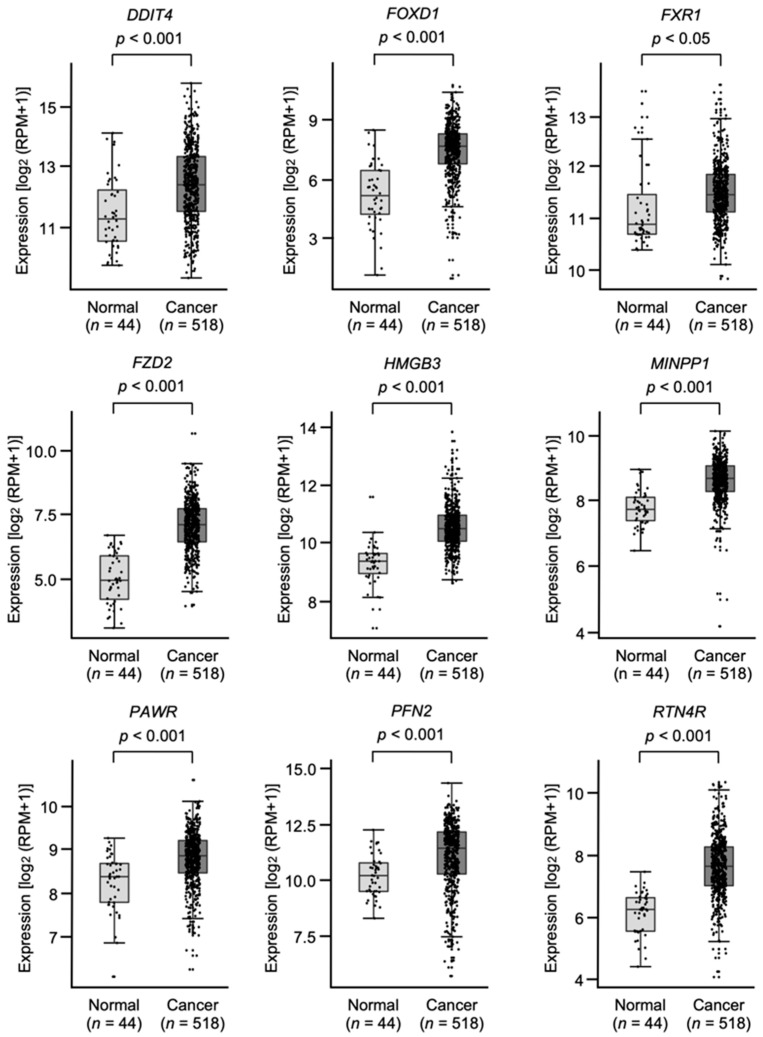
Expression levels of nine target genes (*DDIT4*, *FOXD1*, *FXR1*, *FZD2*, *HMGB3*, *MINPP1*, *PAWR*, *PFN2*, and *RTN4R*) in HNSCC clinical specimens from TCGA-HNSC. All genes were found to be upregulated in HNSCC tissues (*n* = 518) compared with normal tissues (*n* = 44).

**Figure 7 genes-13-01225-f007:**
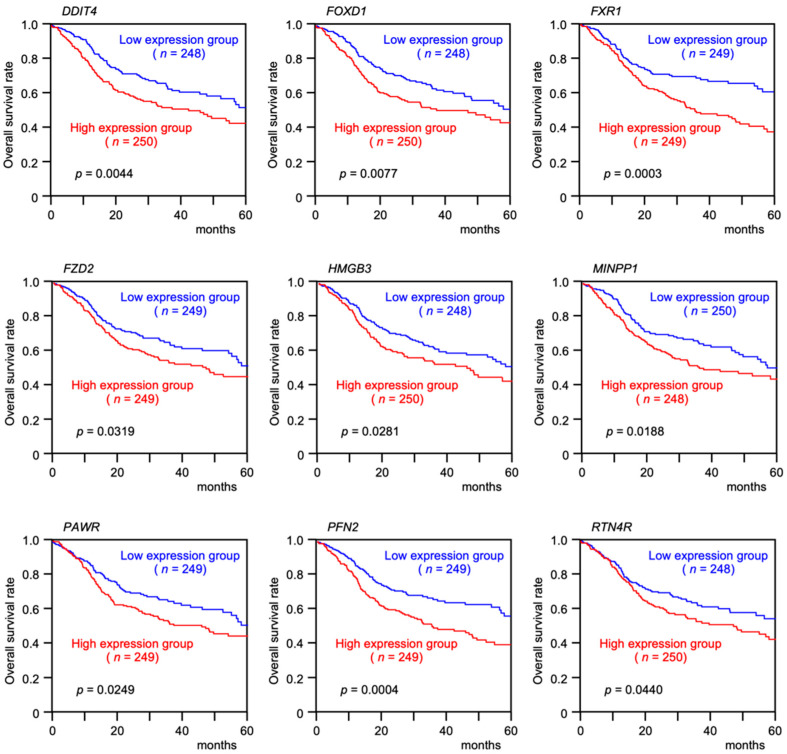
Clinical significance of nine target genes (*DDIT4*, *FOXD1*, *FXR1*, *FZD2*, *HMGB3*, *MINPP1*, *PAWR*, *PFN2*, and *RTN4R*) according to TCGA-HNSC data analysis. Kaplan–Meier curves of the 5-year overall survival rates according to the expression of each gene are presented. High expression levels of all nine genes were significantly predictive of a poorer prognosis in patients with HNSCC. Patients were divided into two groups according to the median gene expression level: high and low expression groups. The red and blue lines represent the high and low expression groups, respectively.

**Figure 8 genes-13-01225-f008:**
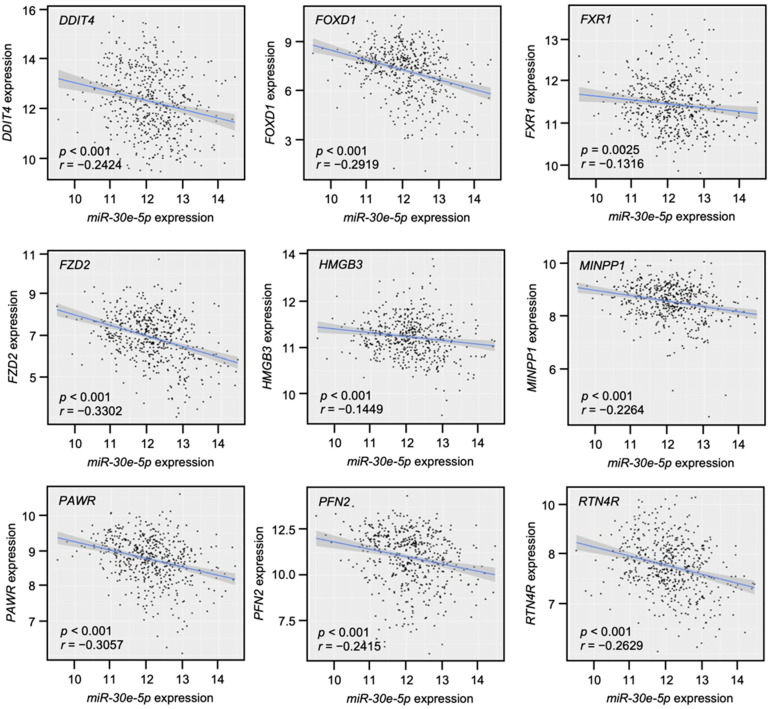
Correlation analysis by TCGA-HNSC for nine genes. Negative correlation of expression levels between *miR-30e-5p* and nine genes in HNSCC clinical specimens.

**Figure 9 genes-13-01225-f009:**
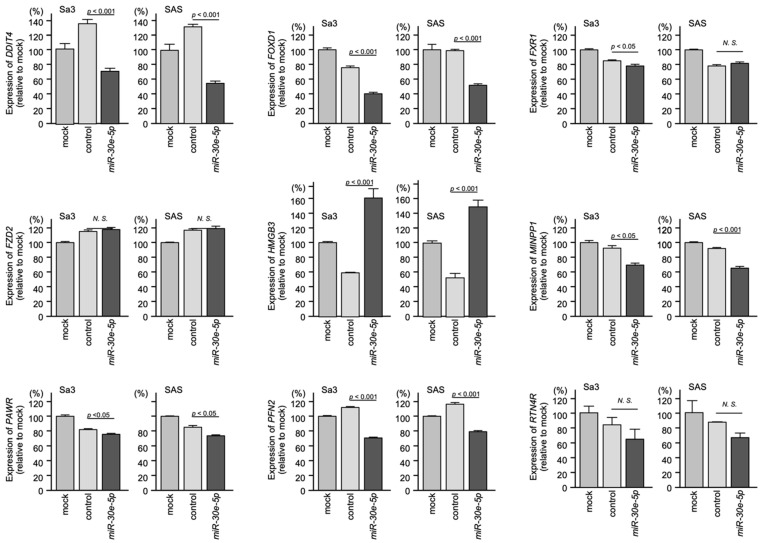
Regulation of the expression of nine genes by *miR-30e-5p* in HNSCC cells. qRT-PCR showing significantly reduced expression of *FOXD1* mRNA (top, middle) 72 h after *miR-30e-5p* transfection in Sa3 and SAS cells (N.S.: not significant compared to control group).

**Figure 10 genes-13-01225-f010:**
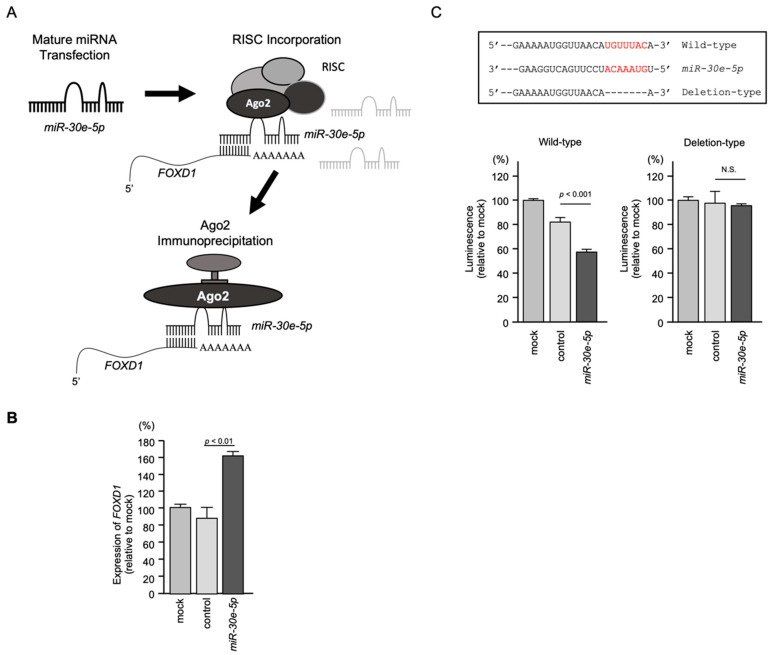
Isolation of RISC-incorporated *FOXD1* mRNA by Ago2 immunoprecipitation. Direct regulation of *FOXD1* expression by *miR-30e-5p* in HNSCC cells. (**A**) Schematic illustration of RIP assay. (**B**) qRT-PCR suggested *FOXD1* mRNA was significantly incorporated into RISC. (**C**) TargetScan database analysis predicting putative *miR-30e-5p*-binding sites in the 3′-UTR of *FOXD1* (upper panel). Dual-luciferase reporter assays showed reduced luminescence activity after co-transfection of the wild-type vector and *miR-30e-5p* in Sa3 cells (lower panel). Normalized data were calculated as the *Renilla/Firefly* luciferase activity ratio (N.S.: not significant compared to control group).

**Figure 11 genes-13-01225-f011:**
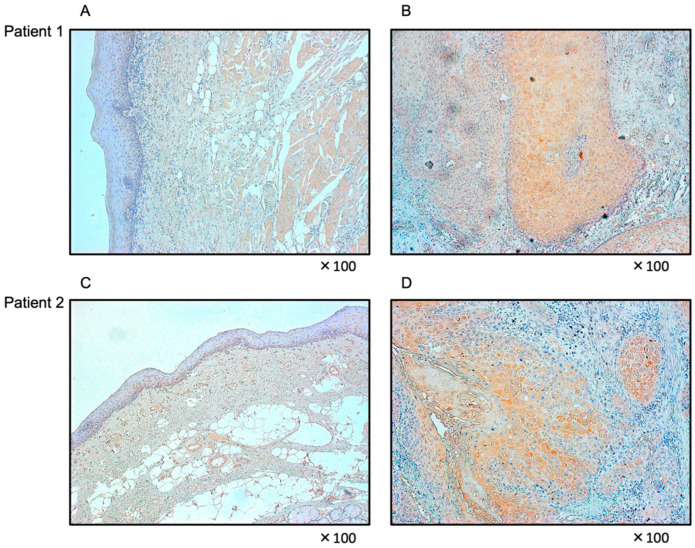
Immunohistochemical staining of *FOXD1* in HNSCC clinical specimens. Weak to moderate immunoreactivity of *FOXD1* was observed in the cancer lesions (**B**,**D**) whereas negative immunoreactivity was shown in normal mucosa (**A**,**C**).

**Figure 12 genes-13-01225-f012:**
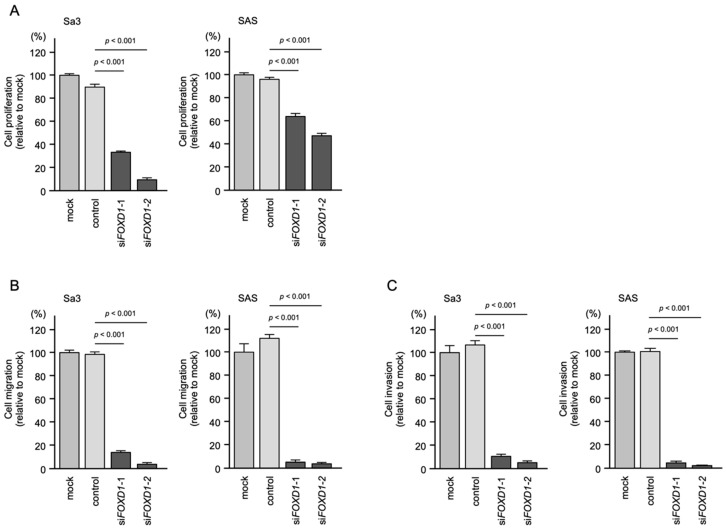
Functional assays of cell proliferation, migration, and invasion following transient transfection of siRNAs targeting *FOXD1* in HNSCC cell lines (Sa3 and SAS cells). (**A**) Cell proliferation assessed by XTT assay 72 h after siRNA transfection. (**B**) Cell migration assessed using a membrane culture system 48 h after seeding miRNA-transfected cells into the chambers. (**C**) Cell invasion assessed by Matrigel invasion assays 48 h after seeding miRNA-transfected cells into chambers.

**Figure 13 genes-13-01225-f013:**
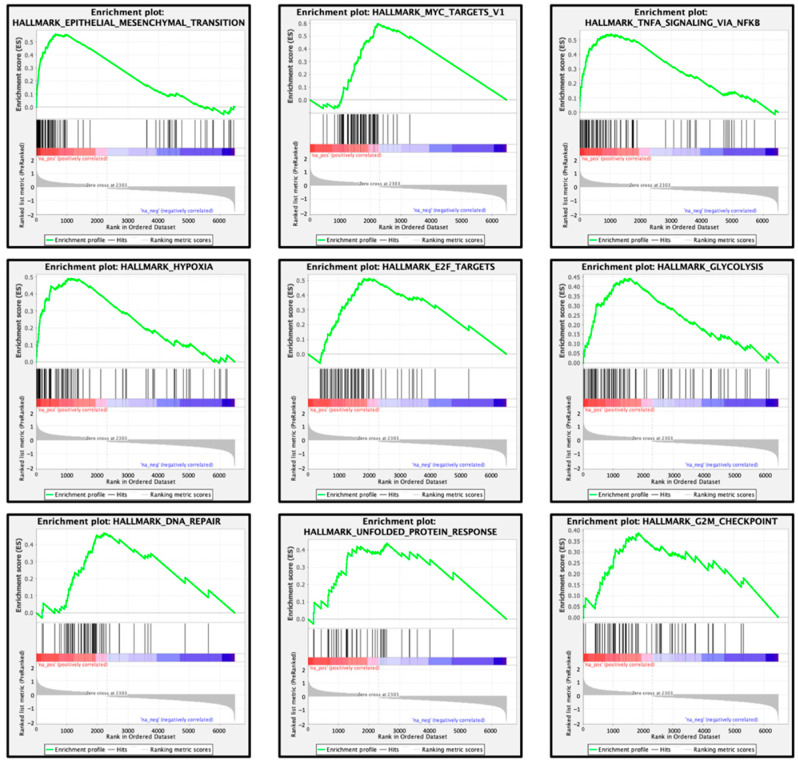
Pathways enriched among the differentially expressed genes in the high *FOXD1* expression group according to gene set enrichment analysis. The nine significantly enriched pathways (top 9) are shown. Most enriched pathway was “Epithelial Mesenchymal Transition”.

**Table 1 genes-13-01225-t001:** Candidate target genes regulated by *miR-30-5p*.

Entrez Gene ID	Gene Symbol	Gene Name	Total Binding Sites	GEO ^1^ *p* Value	GEO log_2_(FC ^2^)	5y OS ^3^ *p* Value
8087	*FXR1*	Fragile X mental retardation, autosomal homolog 1	1	0.014	3.06	<0.001
5217	*PFN2*	Profilin 2	1	0.004	2.67	<0.001
54541	*DDIT4*	DNA-damage-inducible transcript 4	1	0.017	2.70	0.004
2297	*FOXD1*	Forkhead box D1	1	0.008	3.79	0.008
9562	*MINPP1*	Multiple inositol-polyphosphate phosphatase 1	1	0.013	2.02	0.019
5074	*PAWR*	PRKC, apoptosis, WT1, regulator	2	0.003	2.04	0.025
3149	*HMGB3*	High mobility group box 3	1	0.004	2.17	0.028
2535	*FZD2*	Frizzled class receptor 2	1	0.006	2.36	0.032
65078	*RTN4R*	Reticulon 4 receptor	1	0.008	2.20	0.044
115908	*CTHRC1*	Collagen triple helix repeat containing 1	1	0.010	2.96	0.059
3218	*HOXB8*	Homeobox B8	1	0.046	3.29	0.077
9143	*SYNGR3*	Synaptogyrin 3	1	0.042	2.12	0.077
1012	*CDH13*	Cadherin 13	1	0.013	2.13	0.091
6683	*SPAST*	Spastin	2	0.012	2.10	0.092
79718	*TBL1XR1*	Transducin (β)-like 1 X-linked receptor 1	2	0.002	2.63	0.104
114088	*TRIM9*	Tripartite motif containing 9	1	0.005	4.27	0.113
84733	*CBX2*	Chromobox homolog 2	1	0.005	3.00	0.133
27	*ABL2*	ABL proto-oncogene 2, non-receptor tyrosine kinase	1	0.008	2.19	0.155
23657	*SLC7A11*	solute carrier family 7 (Anionic amino acid transporter light chain, xc- system), member 11	1	0.007	4.00	0.160
154214	*RNF217*	Ring finger protein 217	1	0.016	2.36	0.186
79712	*GTDC1*	Glycosyltransferase-like domain containing 1	1	0.004	4.31	0.193
26059	*ERC2*	ELKS/RAB6-interacting/CAST family member 2	1	0.027	3.54	0.210
3237	*HOXD11*	Homeobox D11	1	0.033	4.24	0.214
89796	*NAV1*	Neuron navigator 1	1	0.007	2.88	0.234
6659	*SOX4*	SRY (sex determining region Y)-box 4	1	0.005	2.30	0.258
54434	*SSH1*	Slingshot protein phosphatase 1	1	0.016	2.05	0.280
2048	*EPHB2*	EPH receptor B2	1	0.013	2.53	0.303
9258	*MFHAS1*	Malignant fibrous histiocytoma amplified sequence 1	1	0.005	2.27	0.311
54566	*EPB41L4B*	Erythrocyte membrane protein band 4.1 like 4B	1	0.004	3.06	0.413
8448	*DOC2A*	Double C2-like domains, α	2	0.021	3.00	0.485
28982	*FLVCR1*	Feline leukemia virus subgroup C cellular receptor 1	1	0.005	2.43	0.492
55785	*FGD6*	FYVE, RhoGEF and PH domain containing 6	1	0.014	2.21	0.530
490	*ATP2B1*	ATPase, Ca++ transporting, plasma membrane 1	1	0.007	2.93	0.556
4644	*MYO5A*	Myosin VA (heavy chain 12, myoxin)	1	0.003	2.06	0.605
4015	*LOX*	Lysyl oxidase	1	0.006	4.46	0.613
50805	*IRX4*	Iroquois homeobox 4	1	0.039	2.67	0.652
23432	*GPR161*	G protein-coupled receptor 161	1	0.008	2.50	0.662
2729	*GCLC*	Glutamate-cysteine ligase, catalytic subunit	1	0.004	3.14	0.710
8038	*ADAM12*	ADAM metallopeptidase domain 12	2	0.003	4.29	0.721
3631	*INPP4A*	Inositol polyphosphate-4-phosphatase, type I, 107kDa	1	0.003	2.12	0.747
9832	*JAKMIP2*	Janus kinase and microtubule interacting protein 2	1	0.010	4.19	0.769
121268	*RHEBL1*	Ras homolog enriched in brain like 1	1	0.004	2.92	0.775
144455	*E2F7*	E2F transcription factor 7	3	0.021	2.47	0.786
94032	*CAMK2N2*	Calcium/calmodulin-dependent protein kinase II inhibitor 2	1	0.003	2.95	0.823
84206	*MEX3B*	Mex-3 RNA binding family member B	2	0.016	2.75	0.880
2887	*GRB10*	Growth factor receptor-bound protein 10	1	0.008	2.07	0.890
23333	*DPY19L1*	Dpy-19-like 1 (C. elegans)	1	0.013	3.07	0.893
9435	*CHST2*	Carbohydrate (N-acetylglucosamine-6-O) sulfotransferase 2	1	0.006	3.30	0.917
54165	*DCUN1D1*	DCN1, defective in cullin neddylation 1, domain containing 1	2	0.009	2.50	0.938
54714	*CNGB3*	Cyclic nucleotide gated channel β 3	1	0.008	2.98	0.968
221002	*RASGEF1A*	RasGEF domain family, member 1A	1	0.005	2.81	0.968
55144	*LRRC8D*	Leucine rich repeat containing 8 family, member D	1	0.002	2.60	0.978
8534	*CHST1*	Carbohydrate (keratan sulfate Gal-6) sulfotransferase 1	1	0.005	4.41	0.980
8632	*DNAH17*	Dynein, axonemal, heavy chain 17	1	0.006	5.26	0.999

^1^ Gene Expression Omnibus, ^2^ Fold Change, ^3^ 5-year Overall Survival.

**Table 2 genes-13-01225-t002:** Results of Cox regression analysis of overall survivals in five years in TCGA-HNSC.

	Monovariate			Multivariate		
Variables	HR	95% CI	*p*-Value	HR	95% CI	*p*-Value
*DDIT4* (High vs. Low expression)	1.506	1.138–1.994	0.004	1.292	0.939–1.776	0.115
*FOXD1* (High vs. Low expression)	1.526	1.153–2.019	0.003	1.374	1.002–1.890	0.049
*FXR1* (High vs. Low expression)	1.651	1.242–2.193	0.001	1.303	0.930–1.825	0.124
*FZD2* (High vs. Low expression)	1.337	1.011–1.768	0.042	1.069	0.776–1.473	0.684
*HMGB3* (High vs. Low expression)	1.413	1.069–1.867	0.015	1.186	0.861–1.633	0.297
*MINPP1* (High vs. Low expression)	1.451	1.096–1.921	0.009	1.308	0.940–1.818	0.111
*PAWR* (High vs. Low expression)	1.438	1.084–1.908	0.012	1.255	0.913–1.727	0.162
*PFN2* (High vs. Low expression)	1.642	1.238–2.178	0.001	1.238	0.885–1.731	0.213
*RTN4R* (High vs. Low expression)	1.326	1.003–1.752	0.048	0.962	0.699–1.323	0.810
Age (≥70 vs. <70)	1.628	1.197–2.213	0.002	1.922	1.369–2.698	<0.001
Disease Stage (III, IV vs. I, II)	1.746	1.158–2.633	0.008	1.774	1.159–2.716	0.008
Pathological Grade (3, 4 vs. 1, 2)	0.901	0.653–1.245	0.529	-	-	-

HR: hazard ratio, CI: confidence interval.

**Table 3 genes-13-01225-t003:** Results of gene set enrichment analysis.

Enriched Gene Sets in High *FOXD1* Expression Group		
Name	Normalized Enrichment Score	FDR *q*-Value
Epithelial mesenchymal transition	3.674	*q* < 0.001
*MYC* targets V1	3.412	*q* < 0.001
*TNFα* signaling via NFκB	3.262	*q* < 0.001
Hypoxia	3.044	*q* < 0.001
E2F targets	2.988	*q* < 0.001
Glycolysis	2.797	*q* < 0.001
DNA repair	2.602	*q* < 0.001
Unfolded protein response	2.275	0.001
G2M checkpoint	2.248	0.001
*TGFβ* signaling	2.025	0.001
KRAS signaling up	1.946	0.005
Coagulation	1.848	0.010
Inflammatory response	1.753	0.014
Apical junction	1.739	0.015
Oxidative phosphorylation	1.713	0.015
*P53* pathway	1.702	0.016
*IL6/JAK/STAT3* signaling	1.617	0.027
Apoptosis	1.601	0.029

## Data Availability

Our expression data were deposited in the GEO database (accession number: GSE180077).

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
