# Peer review of "Identification of Antitumor miR-30e-5p Controlled Genes; Diagnostic and Prognostic Biomarkers for Head and Neck Squamous Cell Carcinoma"

_genes, 2022, doi:10.3390/genes13071225_

Round 1
Reviewer 1 Report
The manuscript by Minemura et al., entitled „ Identification of Antitumor miR-30e-5p Controlled Genes; Diagnostic and Prognostic Biomarkers for Head and Neck Squamous Cell Carcinoma“ combines in silico analyses with laboratory approach. It explores the function of the members of miR-30 family in head and neck squamouse cell carcinomas.
Specific comments:
The manuscript is well written and in the study a wide variety of methods is utilized which allows for complex analyses of the function of miRNAs from the miR-30 family and selected target genes including the survival analyses.
I have few minor comments.
Numerous studies have shown the large heterogeneity in the miRNAs expression in tissues of different anatomical location. The wet laboratory part of the work has been done only on tissues from the hypopharyngeal region, and only two samples were analysed. It would be interesting to see if the results are also valid for HNSCC of other anatomical locations and for more samples.
In the Materials and Methods, Statistical analyses: the variables included in the Cox´s proportional HR should be listed.
For FXR1 gene, the prognostic p value was 0.052 that is not statistically significant.
Reviewer 2 Report
In the manuscript by Minemura et al the authors descriibe a study of miR-30e regulated genes in the context of head and neck cancer. The study is robust employing bioinformatical assessment, patient survival, RNA pulldowns, qPCR, IHC in clinical samples, si silencing, luciferase assays, as well as the usual functional tests (migration, invasion, proliferation) in cell lines. It is more or less the same study as the previous for miR-30e-3p
Possibly due the complexity of the manuscript, some methodological descriptions are not as comprehensive as they should be.
Some of the issues are listed below:
P3 L 101 was some database of interactions also assessed beyond computational target prediction using targetscan? For example TarBase supposedly collects the data on all validated miRNA:target gene interactions (https://dianalab.e-ce.uth.gr/html/diana/web/index.php?r=tarbasev8%2Findex) which might be a way to find more interesting targets with experimental support
P3L104 why were hypopharyngeal cancer cases selected? Head and neck cancer is a diverse disease and hypopharyngeal cancer is not that representative of HNSCC as a whole (https://www.ncbi.nlm.nih.gov/books/NBK567720/). Its proportion is a higher in Asia (so this might be highlighted as support of the choice) but still it should be somehow justified in the text
P3 L 106 it is unclear how many patients were enrolled, what kind of tissue was collected and how was the material processed. No patient data (or inclusion/exclusion criteria) is shown at this point. When looking at the related results section, no results about patient sample microarray analysis is shown in this subsection 3.1. Either the methods are poorly written or the manuscript should be restructured so that the methods and results flow in the same order.
P3L114 The supplementary table 1 suggests that the 2 cell lines used are derived from upper gingiva and tongue cancer cases. Since the table is relatively small, this data could fit in the main text easily. Furthermore it is incomplete since cellosaurus provides the missing data (age, gender) https://web.expasy.org/cellosaurus/CVCL_1675 if this is the same “SAS” cell line? It might be worth noting that the cell line is derived from oral tongue so it is not confused with base of tongue (since Base of tongue site is more likely to have HPV involvement, while oral tongue is not)
P3 L137-138 since transfection is mentioned at line 127 the reagents description (if the same) could be moved to that earlier point
P4170-173 The patients are inconsistent from above? For IHC the authors used 2 cases of (oral?) tongue cancers (typo in one), while previously hypopharyngeal cancer patiens were selected for microarray analysis? Why the same patients were not assessed by both methods? HNSCC is a diverse set, adding further noise by mismatching sample subsites might further complicate matters.
P4 L172 since only 1 antibody was used, it might be more appropriate to list the information in the main text.
P6 L219 The survival curves (Figure 3) suggest there were 251 patients in low and 251 patients in miR-30a-5p high group. Yet the figure 2 suggests there was a total of 484 cases with miRNA data which is impossible? Similar is with Figs 6 and 7 which list again different numbers of patients but at least there is less patients with survival data than with expression data as expected. But please double check the groups assessed.
P6L 229-235 The phrase “ectopic expression” might be inappropriate since the miRNA was not actually expressed from some plasmid but was simply transfected.
P7 L250 the subheading is misleading since “HNSCC cells” suggests cell lines while the data is from cancer patients. The methodology about how GSE180077 was made is still missing at this point of the manuscript. At least a reference to previous publication is necessary as well as inclusion of some relevant data (numbers and types of samples). What targets would be obtained if only TCGA data was considered without needing to overlap with your hypopharyngeal results? How consistent are your results to the TCGA subset of only hypopharyngeal cancer cases? Was any validation done for GSE180077 results?
Table 1 Row spacing could be condensed so that it doesn’t stretch across 3 pages
P11 L287 how was the multivariate Cox model made (enter, forward, backward, stepwise)? From the figure 8 it appears that all variables were used and retained in the model (= enter option) while It might be more appropriate to first see which variables are worth including and then construct a model with only those (ie. forward option)
P12 L297 please specify the dataset used for correlation analysis
P18 L434 the paragraph about profilins seems misplaced since PFN2 was not particularly highlighted in the results
P19 L443 the paragraph about FOXD1 should be earlier in the discussion since this is the main highlighted result.
P3 L136 Typo “26]. T The”
